# Inverse Online Learning: Understanding Non-Stationary and Reactionary Policies

**Alex J. Chan, Alicia Curth, and Mihaela van der Schaar**
University of Cambridge
Department of Applied Mathematics and Theoretical Physics
Cambridge, UK
`{ajc340, amc253, mv472}@cam.ac.uk`

## Abstract

Human decision making is well known to be imperfect and the ability to analyse such processes individually is crucial when attempting to aid or improve a decision-maker's ability to perform a task, e.g. to alert them to potential biases or oversights on their part. To do so, it is necessary to develop interpretable representations of how agents make decisions and how this process changes over time as the agent *learns online* in reaction to the accrued experience. To then understand the decision-making processes underlying a set of observed trajectories, we cast the policy inference problem as the inverse to this online learning problem. By interpreting actions within a *potential outcomes* framework, we introduce a meaningful mapping based on agents choosing an action they *believe* to have the greatest treatment effect. We introduce a practical algorithm for retrospectively estimating such perceived effects, alongside the process through which agents update them, using a novel architecture built upon an expressive family of deep state-space models. Through application to the analysis of UNOS organ donation acceptance decisions, we demonstrate that our approach can bring valuable insights into the factors that govern decision processes and how they change over time.

## 1 Introduction

Decision modelling is often viewed through the lens of policy learning (Bain & Sammut, 1995; Abbeel & Ng, 2004), where the aim is to learn some imitation policy that captures the actions (and thus decisions) of an agent in some structured way (Hüyük et al., 2021a). Unfortunately, this usually implicitly relies on the assumption that the actions taken are already *optimal* with respect to some objective, and hence do not change over time. While this might be appropriate to approximate the stationary policy of an autonomous agent, such assumptions often break down when applying them to more flexible agents that might be learning on-the-fly. In particular, we may want to model the decision making process of a human decision-maker, whose actions we may see executed over time - this is clearly of great use in a range of fields including economics and medicine (Hunink et al., 2014; Zavadskas & Turskis, 2011). In the context of machine learning, they would be considered to be undergoing a process of *online learning* (Hoi et al., 2018); training while interacting with the environment, where it is to be expected that beliefs and strategies are constantly adapted based on accumulating experience (Elio & Pelletier, 1997). Additionally, such belief updates are often imperfect; for example, people are known to *a priori* over-weight the potential impact of rare events when they first start a task, before proceeding to neglect them when they do not occur in their experience (Hertwig & Erev, 2009). If our goal is to analyse human decision-making, it is thus important to be able to model non-stationary policies that are *potentially* imperfect in two senses: both marginally at each time-step, and in the way that they become adapted to new information.

In this work, instead of relying on the assumption that an agent follows a stationary optimal policy, we consider their actions to be consistent within an *online leaning* framework where they learn from an incoming sequence of examples (Hoi et al., 2018) and update their policies accordingly, as in Figure 1. Thus, instead of assuming that actions are globally optimal, we assume that at any *exact time* the agent *thought* that the action was *better* than the alternatives. As a natural consequence, we frame the problem of decision modelling as that of *inverse* online learning; that is, we aim to uncover

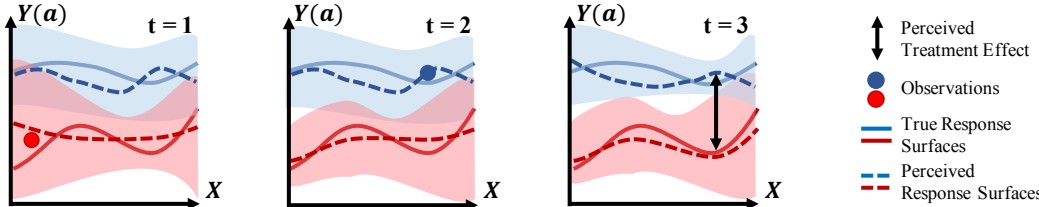

Figure 1: **Capturing online learning behaviour.** We assume the agent models the response surfaces in their mind, and that their belief changes each time step in response to the actions they take and the consequent observations they make. The response surface they model need not converge on the true surface since the update method of the agent may be biased and not appropriately use the information available to them.

the process through which an agent is learning online. Note that this does not involve coming up with a new way to do online learning itself, rather we recover how it appears a given agent may have learnt. We do this by modelling the (non-stationary) policy of the agent using a deep state-space model (Krishnan et al., 2017), allowing us to track the *memory* of the agent while allowing policies to evolve in a non-parametric and non-prescribed fashion. We construct an interpretable mapping from the accrued memory to policies by framing the decision-making process within a potential-outcome setting (Rubin, 2005); letting actions correspond to interventions (or *treatments*), and assuming that agents make decisions by choosing an action which they *perceive* to have the highest (treatment) effect.

**Contributions.** In this work we make a number of contributions: First, we formalise the inverse online learning problem and a connection between decision modelling, online learning and potential outcomes, highlighting how they complement each other to result in an interpretable understanding of policies (Section 2); Second, we introduce a practical method to estimate the non-stationary policy and update method of an agent given demonstrations (Section 4); And third, we demonstrate how we can uncover useful practical insights into the decision making process of agents with a case study on the acceptance of liver donation offers (Section 5). Code is made available at `https://github.com/XanderJC/inverse-online`, along with the group codebase at `https://github.com/vanderschaarlab/mlforhealthlabpub`.

## 2 PROBLEM FORMALISATION

**Preliminaries.** Assume we observe a data trajectory[1] of the form $\mathcal{D} = \{(X_t, A_t, Y_t)\}_{t=1}^T$. Here, $X_t \in \mathcal{X} \subset \mathbb{R}^d$, denotes a context vector of possible confounders; $A_t \in \mathcal{A} = \{0,1\}$, a binary action or intervention; and $Y_t \in \mathcal{Y}$, a binary or continuous outcome of interest. The subscript $t$ indicates a time-ordering of observed triplets; any time-step $t$ is generated by (i) arrival of context $x$, (ii) an intervention $a$ being performed by an agent according to some policy $\pi_t(x)$ and (iii) a corresponding outcome $y$ being observed.

**Goal.** We are interested in recovering the agent's *non-stationary* policy $\pi_t(x) = P(A = 1|X = x, H_t = h_t)$, which depends on the observed context $x$ and can change over time due to the observed history $H_t = \{(X_k, A_k, Y_k)\}_{k=1}^{t-1}$. We make the key assumption that the agent acts with the *intention* to choose an intervention $a$ leading to the best (largest) potential outcome $Y(a)$, but that the *perception* of the optimal intervention may change over time throughout the agent's *learning process* about their environment. This problem formalisation naturally places us at the in-

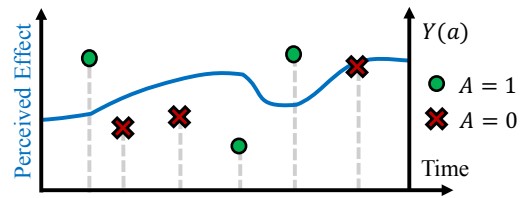

Figure 2: **Simplified inverse process.** Our task is to infer the evolving perceived effect over time (the blue line) given only realisations of the agent's behaviour.

tersection of treatment effect inference within the Neyman-Rubin potential outcomes (PO) framework

---

[1]We assume with a single trajectory here, though learning and inference can be applied over a dataset of multiple trajectories

(Rubin, 2005) with the classical (inverse) reinforcement learning (RL) paradigm (Abbeel & Ng, 2004). More specifically though the RL-subproblem of logged contextual bandits (Swaminathan & Joachims, 2015; Atan et al., 2018), where the setting is offline and the contexts arrive independently.

**Perception Versus Reality.** As is standard within the PO framework, we assume that any observation has two potential outcomes $Y(a)$, associated with each intervention (treatment-level), which are generated from some true distribution inherent to the environment, i.e. $Y(0), Y(1) \sim \mathbb{P}^e(\cdot|X = x)$, yet only the potential outcome associated with the chosen treatment is revealed to the agent. If the agent had full knowledge of the environment, its optimal policy would be to choose an action as $\arg \max_a \mu_a^e(x)$ where $\mu_a^e(x) = \mathbb{E}_{\mathbb{P}^e}[Y(a)|X = x]$, or conversely to choose to perform the intervention only when its expected conditional average treatment effect (CATE) $\tau^e(x) = \mu_1^e(x) - \mu_0^e(x)$ is positive. In general, we assume that such perfect knowledge of the environment is not available to the agent in many scenarios of practical interest. Instead, we assume that at any time $t$, the agent acts according to some *perceived* model $\tilde{\mathbb{P}}$ of its environment, which induces the perceived expected potential outcomes $\tilde{\mu}_a(x; t)$ and associated treatment effect $\tilde{\tau}(x; t)$. Here, $\tilde{\mu}_a(x; t) = \mathbb{E}_{\tilde{\mathbb{P}}}[Y(a)|X = x, H_t = h_t]$, is a function not only of the observed context but also the history, such that the agent can update its beliefs according to observed outcomes. Note that we allow the agent's model of the environment to be misspecified; in this case $\tilde{\mu}_a(x; t)$ would not converge to $\mu_a^e(x)$ even for very long trajectories. We consider this an important feature, as we assume that human decision-makers generally need to make simplifications (Griffiths et al., 2015). Formally, we make the following assumption about the agent's decision making process:

**Assumption 1** *Mutual Observability: The observable space $\mathcal{X}$ contains only and all the information available to the agent at the point of assigning an action.*

**Assumption 2** *Perceived Optimality: An agent assigns the action they think will maximise the outcome. For a deterministic policy $\pi_t$, the agent believes $\pi_t(x) = 1 \iff \tilde{\mu}_1(x; t) > \tilde{\mu}_0(x; t)$. Given a stochastic agent, the policy $\pi_t(x) = P(A_t = 1|x)$ is a monotonic increasing function of the agent's belief over the treatment effect $\tilde{\tau}(x; t)$.*

**Assumption 3** *Continual Adaptation: The agent may continually adjust their strategy based on experience. The index in the dataset $\mathcal{D}$ represents a temporal ordering which $\pi$ is non-stationary with respect to, i.e. $\pi_i(x)$ may not equal $\pi_j(x), i \neq j \geq 1$ where $\pi_i$ represents the policy at time step $i$.*

Assumptions 1 and 2 are generally unverifiable in practice, yet, they are necessary: there would be little opportunity to learn anything meaningful about a non-stationary policy without them. Mutual observability is crucial, as assuming that different information streams are available to the observer and the agent would make it impossible to accurately describe the decision-making process. Assumption 2, albeit untestable, appears very reasonable: in the medical setting, for example, a violation would imply malicious intent – i.e. that a clinician purposefully chooses the intervention with a sub-optimal expected outcome. Nonetheless, this does mean that we *do not* consider exploratory agents that are routinely willing to sacrifice outcomes for knowledge, a constraint we do not consider too restrictive within the medical applications we have in mind. Assumption 3, on the other hand, is not at all restrictive; on the contrary, it explicitly specifies a more flexible framework than is usual in imitation or inverse reinforcement learning as the set of policies it describes contains stationary policies.

## 3 RELATED WORK

Our problem setting is related to, but distinct from, a number of problems considered within related literature. Effectively, we assume that we observe *logged* data generated by a learning agent that acts according to its evolving belief over effects of its actions, and aim to solve the inverse problem by making inferences over said belief given the data. Below, we discuss our relationship to work studying both the forward and the inverse problem of decision modelling, with main points summarised in Table 1.

**The Forward Problem: Inferring *Prescriptive* Models for Behaviour.** The agent, whose (non-stationary) policy we aim to understand, actively learns *online* through repeated interaction with an environment and thus effectively solves an online learning problem (Hoi et al., 2018). The most prominent example of such an agent would be a *contextual bandit* (Agarwal et al., 2014), which learns to assign sequentially arriving contexts to (treatment) arms. Designing agents that update

policies online is thus the *inverse* of the problem we consider here. Another seemingly related problem within the RL context is learning *optimal* policies from logged bandit data (Swaminathan & Joachims, 2015); this is different from our setting as it is *prescriptive* rather than *descriptive*.

If the goal was to infer the true $\tau^e(x)$ from the (logged) observational data $\mathcal{D}$ (instead of the effect *perceived* by the agent), this would constitute a standard static CATE estimation problem, for which many solutions have been proposed in the recent ML literature (Alaa & van der Schaar, 2018; Shalit et al., 2017; Künzel et al., 2019). Note that within this literature, the treatment-assignment policy (the so-called propensity score $\pi(x) = \mathbb{P}(A = 1 | X = x)$) is usually assumed to be stationary and considered a *nuisance* parameter that is not of primary interest. Estimating CATE can also be a pretext task to the problem of developing *optimal treatment rules* (OTR) (Zhang et al., 2012a;b; Zhao et al., 2012), as $g(x) = \mathbb{1}(\tau(x) > 0)$ is such an optimal rule (Zhang et al., 2012b).

**The Backward Problem: Obtaining a *Descriptive* Summary of an Agent.** When no outcomes are observed (unlike in the bandit problem) the task of learning a policy becomes imitation learning (IL), itself a large field that aims to match the policy of a demonstrator (Bain & Sammut, 1995; Ho & Ermon, 2016). Note that, compared to the bandit problem, there is a subtle shift from obtaining a policy that is optimal with respect to some *true* notion of outcome to one that minimises some divergence from the demonstrator. While IL is often used in the forward problem to find an optimal policy – thereby implicitly assuming that the demonstrator is themselves acting optimally –, it can also, if done interpretably, be used in the backward problem to very effectively reason about the goals and preferences of the agent (Hüyük et al., 2021a). One way to achieve this is through inverse reinforcement learning (IRL), which aims to recover the reward function that is seemingly maximised by the agent (Ziebart et al., 2008; Fu et al., 2018; Chan & van der Schaar, 2021). This need not be the *true* reward function, and can be interpreted as a potentially more compact way to describe a policy, which is also more easily portable given shifts in environment dynamics.

Our problem formalisation is conceptually closely related to the *counterfactual* inverse reinforcement learning (CIRL) work of Bica et al. (2021). There the authors similarly aim to explain decisions based on the PO framework, specifically by augmenting a max-margin IRL (Abbeel & Ng, 2004) approach to parameterise the learnt reward as a weighted sum of counterfactual outcomes. However, CIRL involves a pre-processing counterfactual estimation step that focuses on estimating the *true* treatment effects, and implicitly assumes that these are identical to the ones *perceived* by the agent (i.e. it assumes that the agent has perfect knowledge of the environment dynamics generating the potential outcomes), marking a significant departure from our (much weaker) assumptions. Without the focus on treatment effects, our method could be seen as a generalised non-parametric extension of the inverse contextual bandits of Hüyük et al. (2021b). In general, the backward problem is hard to evaluate empirically since information about the true beliefs of agents is not normally available in the data, thus relying on simulation to validate (Chan et al., 2021).

Table 1: Problems considered in related work, noting the relationship between both input and output.

| Problem | Input | Target quantity |
|---|---|---|
| Online Learning of Optimal Polices (Agarwal et al., 2014) | $(X_t, A_t, Y_t)$ | $\pi_t^{opt}(x)$ |
| Learning Optimal Policy from Logged Bandits (Swaminathan & Joachims, 2015) | $(X_i, A_i, Y_i)$ | $\pi^{opt}(x)$ |
| Estimating Heterogeneous Treatment Effects (Alaa & van der Schaar, 2018) | $(X_i, A_i, Y_i)$ | $\tau^e(x)$ |
| Learning Imitator Policies via Imitation Learning (Bain & Sammut, 1995) | $(X_i, A_i)$ | $\pi^{obs}(x)$ |
| Learning Reward Functions via IRL (Abbeel & Ng, 2004) | $(X_i, A_i)$ | $R^{obs}(x)$ |
| Learning Reward Functions via CIRL (Bica et al., 2021) | $(X_i, A_i, Y_i)$ | $R^{obs}(x)$ |
| Modelling Dynamic Policies via Inverse Online Learning | $(X_t, A_t, Y_t)$ | $\pi_t^{obs}(x)$ |

## 4 Inferring the Online Learning Process

**Preliminaries: The Forward Problem.** In the forward context, the problem we consider amounts to an agent attempting to, at each time-step, take the action they believe will maximise the outcome, without being aware of the *true* effect of their intervention. We assume that the agent *believes* the potential outcomes are a linear combination of the contextual features such that $\tilde{\mu}_a(X_t; t) = \langle X_t, \omega_a^t \rangle$, with $\omega_a^t \in \Omega$ a set of weights for the action $a$ at time $t$, and $\langle \cdot, \cdot \rangle$ denoting the inner product. Normally we may consider a linear model an oversimplification, but

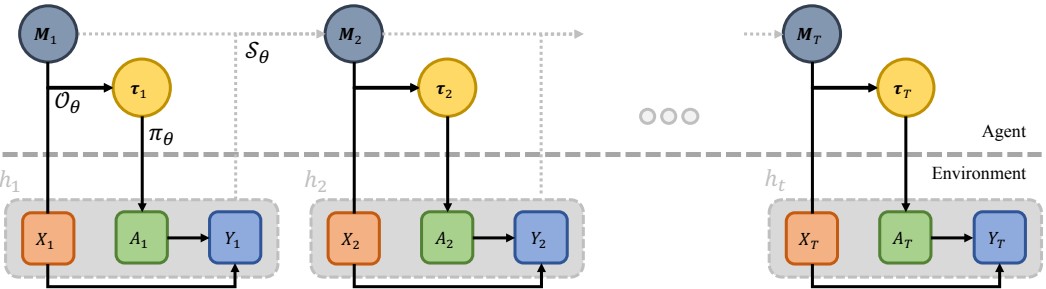

Figure 3: **Model.** Solid black lines indicate single time step dependencies. Given covariates $X_i$ and an observed history then treatment effects are estimated in the mind of the agent. An action $A_i$ is selected based on these effect and an outcome $Y_i$ is observed. The dashed grey lines then indicate information flow over time.

when modelling the thought process of an agent it appears much more reasonable - for example, does a doctor consider higher order terms when weighing up potential outcomes of a patient? It seems unlikely, especially as clinical guidelines are often given as linear cutoffs (Burgers et al., 2003). The considered forward problem thus proceeds as follows for every given time step $t$:

1. A context $X_t \sim P_\mathcal{X}$ arrives independently over time. We make no assumptions on the form of $P_\mathcal{X}$, in particular noting that there is no need for any level of stationarity.

2. Given $X_t$ and the current set of weights $\{\omega_1^t, \omega_0^t\}$, the agent predicts their perceived potential outcomes $\tilde{\mu}_1(X_t; t)$ and $\tilde{\mu}_0(X_t; t)$.

3. Action $A_t$ is assigned by some probabilistic optimal treatment rule $\pi : \mathcal{Y}^2 \mapsto [0, 1]$ such that $\mathbb{P}(A_t = 1|X_t) = \pi(\tilde{\mu}_1(X_t; t), \tilde{\mu}_0(X_t; t))$, before the associated true outcome is observed.

4. Based on this new information the agent (potentially stochastically) updates their belief over the response surface for the taken action according to some update function $f : \Omega^2 \times \mathcal{X} \times \mathcal{A} \times \mathcal{Y} \mapsto \triangle(\Omega^2)$ such that $\omega_1^{t+1}, \omega_0^{t+1} \sim f(\omega_1^t, \omega_0^t, X_t, A_t, Y_t)$.

To *maximise outcomes* it is crucial to employ an update function $f$ that maximally captures the available information at each step of the process. This then allows for an appropriate determination of $\pi$ that aims to maximise the total (potentially future discounted) outcomes.

### 4.1 A Model of Inverse Online Learning

Here, we focus on the *inverse* to the problem outlined above, which, to the best of our knowledge, has received little to no attention in related work. That is, after observing an agent's actions, can we recover how they arrived at their decisions, and how this changed over time? In the following, we use vector notation $\vec{x}_{t:t'} = \{x_i\}_{i=t}^{t'}$ with $\vec{x} = \vec{x}_{1:T}$ for all realised variables, with histories $\vec{h}_{t:t'} = \{(x_i, a_i, y_i)\}_{i=t}^{t'}$. Note that the previously described forward model assumes a factorisation of the full generative distribution of the observed data given by:

$$p(\vec{x}, \vec{a}, \vec{y}) = \prod_{t=1}^{T} p(y_t|a_t, x_t) \times p(a_t|\vec{x}_{1:t-1}, \vec{a}_{1:t-1}, \vec{y}_{1:t-1}) \times p(x_t). \tag{1}$$

We assume that $p(x_t)$ and $p(y_t|a_t, x_t)$ are governed by some true environment dynamics, independent from the perception of the agent and thus not part of our modelling problem. This seems trivially true, as the perception of the agent will neither affect the contexts that arrive nor the true outcome conditional on the action being taken.

Our model for the evolving beliefs of an agent with respect to their interaction with the environment revolves around a specialised deep state-space model. Crucially, this takes the form of a latent random variable $M_t \in \mathbb{R}^d$, which captures the *memory* of the agent at time $t$, and can evolve in a flexible way based on the observed history. We consider the memory to be an efficient summary of the history, which can be interpreted as a sufficient statistic for the beliefs of the agent at a given time. Naturally, this memory is unobserved and must be learnt in an unsupervised

manner. The introduction of this latent variable leads to an extended probabilistic model with a more structured factorisation of the conditional distribution of the actions, and a model given by:

$$p_\theta(\vec{a}|\vec{x}, \vec{y}) = \prod_{t=1}^{T} \underbrace{\pi_\theta(a_t|\tilde{\tau}_t)}_{\textbf{Treatment Rule}} \times \underbrace{\mathcal{O}_\theta(\tilde{\tau}_t|x_t, m_t)}_{\textbf{Outcome Estimation}} \times \underbrace{\mathcal{S}_\theta(m_t|\vec{x}_{1:t-1}, \vec{a}_{1:t-1}, \vec{y}_{1:t-1})}_{\textbf{Memory Aggregation}}. \quad (2)$$

This model consists of three components: a memory summarisation network $\mathcal{S}_\theta : \mathcal{H} \mapsto \triangle(\mathcal{M})$, which maps the history into a (distribution over) memory; a (perceived) potential outcome predictor network $\mathcal{O}_\theta : \mathcal{X} \times \mathcal{M} \mapsto \mathbb{R}^2$, that takes the memory and a context and predicts the perceived potential outcomes; and a treatment rule $\pi_\theta : \mathcal{Y}^2 \mapsto [0, 1]$, that given the potential outcomes (summarised by the perceived treatment effect) outputs a distribution over actions. Throughout, $\theta$ denotes the full set of parameters of the generative model and notation is shared across components to reflect that they can all be optimised jointly. Below, we discuss each component in turn.

**Memory Aggregation.** We define the base structure of $\mathcal{S}_\theta$ to be relatively simple and recursive, and assume the memory at a given time step to be distributed as:

$$M_t \sim \mathcal{N}(\boldsymbol{\mu}_t, \boldsymbol{\Sigma}_t) \quad \text{with} \quad \boldsymbol{\mu}_t, \boldsymbol{\Sigma}_t = \text{MLP}([M_{t-1}, X_{t-1}, A_{t-1}, Y_{t-1}]), \quad (3)$$

where $\boldsymbol{\mu}_t$ and $\boldsymbol{\Sigma}_t$ are two output heads of the memory network $\mathcal{S}$ given the history $h_{t-1}$. It is important to note that the memory network does not get the $t$-th step information (including the context) when predicting the memory at time $t$ - this means the network cannot provide any specific predictive power for a given context and is forced to model the general response surfaces.

In many models of behaviour the update function $f$ of the agent is modelled as perfect Bayesian inference (Kaelbling et al., 1998; Colombo & Hartmann, 2017). While this may seems natural on the surface given the logical consistency of Bayesian inference, it seems unlikely in practice that agents (and especially humans) will be capable of it. Not least because in all but the simplest cases it will become intractable for computational agents and humans themselves are well known to make simplifying approximations when dealing with past histories (Genewein et al., 2015). By employing a flexible neural architecture, we remove this relatively restrictive assumption on the memory update process. Our model then learns exactly how the agent appears to use previous information to update their predictions - allowing us to model common cognitive biases such as overly weighting recent or largely negative events (Hertwig & Erev, 2009).

**Outcome Estimation.** Having obtained a memory state $m_t$, the second stage involves predicting the perceived potential outcomes for a context $x_t$ with the network $\mathcal{O}_\theta(\tilde{\tau}_t|x_t, m_t)$. Given the forward model where the potential outcomes are considered a linear combination of the features this then proceeds again in three steps:

$$\omega_1^t, \omega_0^t = \text{MLP}(m_t), \quad (4)$$

$$\tilde{\mu}_1(x_t; t) = \langle x_t, \omega_1^t \rangle, \quad \tilde{\mu}_0(x_t; t) = \langle x_t, \omega_0^t \rangle \quad (5)$$

$$\tilde{\tau}_t = \tilde{\mu}_1(x_t; t) - \tilde{\mu}_0(x_t; t). \quad (6)$$

First the memory $m_t$ is decoded into a set of weights for each action using some standard feed-forward multi-layer perceptron (MLP), before the potential outcomes are predicted by taking the inner product with the context. This is preferable to, for example, concatenating and passing both $x_t$ and $m_t$ through a network since, while it may be potentially more expressive, it loses the connection to the forward model as well as the interpretability that is given by linearity. As it is, while our model of the potential outcomes is linear *at each time step*, the memory allows this to flexibly change throughout the course of the history. The perceived treatment effect is then calculated as the difference between potential outcomes. However, given the linear nature of the predictions, there is no non-degenerate solution for the $\omega$s. Thus, to ensure identifiability, we set $\omega_0^t = \mathbf{0}$ as a baseline.

**Treatment Rule.** Under Assumption 2, the agent should be expected to take an action that maximises the outcome. There are different possible options for the exact treatment rule, the most obvious being $\pi(a_t|\tilde{\tau}_t) = \mathbb{1}(\tilde{\tau}_t > 0)$; assuming that as long as the treatment effect is positive then the intervention will be taken. However, in order to maintain differentiability, as well as to allow for more modelling flexibility to capture the stochasitcity of agents, we parameterise the policy as a soft version of the indicator function:

$$\pi(a_t|\tilde{\tau}_t) = \frac{1}{1 + \exp(-\alpha(\tilde{\tau}_t - \beta))}. \quad (7)$$

Assuming positive $\alpha$, this is clearly monotonically increasing, satisfying Assumption 2. These parameters are learnt by the model and thus allow us to model a flexible threshold with $\beta$ as well as aleatoric uncertainty in the actions of the agent with $\alpha$.

## 4.2 LEARNING WITH AN INFERENCE NETWORK

As with all flavours of deep state space models, exact posterior inference is analytically intractable, making optimisation of the marginal likelihood equally so. As such, we aim to maximise an Evidence Lower BOund (ELBO) on this likelihood of the data following standardised variational principles (Blei et al., 2017; Hoffman et al., 2013). This is achieved by positing a parameterised *approximate* posterior distribution $q_\phi(\vec{m}|\vec{h})$, and with a simple application of Jensen's inequality arrive at:

$$\log p_\theta(\vec{h}) \geq \mathbb{E}_{q_\phi}\big[\log p_\theta(\vec{m}, \vec{h}) - \log q_\phi(\vec{m}|\vec{h})\big]. \tag{8}$$

Maximising this bound is not only useful in terms of the marginal likelihood but also equivalently minimises the Kullback-Leibler divergence between the approximate and true posterior $D_{KL}[q_\phi(\vec{m}|\vec{h})||p(\vec{m}|\vec{h})]$, giving us more confidence in our inferred posterior.

**Factorising the Posterior for Efficient Inference.**
In order to accelerate learning and make inference quick, we factorise the approximate posterior in the same way as the true posterior:

$$q_\phi(\vec{m}|\vec{h}) = q_\phi(m_1|\vec{h}) \prod_{t=2}^{T} q_\phi(m_t|m_{t-1}, \vec{h}_{t-1:T})$$

$$(9)$$

Thus to estimate the approximate posterior, we use a backwards LSTM to create a summary $b_t = \text{LSTM}(h_{t:T})$. This is then concatenated with the previous memory at time step $t-1$ before being passed through a fully connected layer to estimate parameters. Given that the memory can be seen as a sufficient statistic of the past within the model,

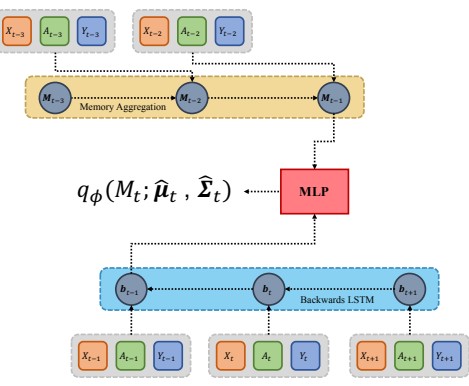

Figure 4: Structure of the approximate posterior inference network.

it is well known that given this factorisation, conditioning on the memory leads to a Rao-Blackwellization of the more general network that would be conditioned on all variables (Krishnan et al., 2017; Alaa & van der Schaar, 2019). This accelerates learning not only by parameter sharing between $\phi$ and $\theta$ but also by reducing the variance of the gradient estimates during training. With the specified inference network $q_\phi$, this leads to an optimisation objective:

$$\mathcal{F}(\phi, \theta) = \mathbb{E}_{q_\phi(\vec{m}|\vec{h})}\big[p_\theta(\vec{a}|\vec{h})\big] - D_{KL}\big(q_\phi(m_1|\vec{x})||p_\theta(m_1)\big) \tag{10}$$

$$- \sum_{t=2}^{T} \mathbb{E}_{q_\phi(m_{t-1}|\vec{h})}\big[D_{KL}\big(q_\phi(m_t|m_{t-1}, \vec{h}_{t-1:T})||p_\theta(m_t|m_{t-1})\big)\big]$$

This can optimised in a straightforward manner using stochastic variational inference and reparameterised Monte Carlo samples from the approximate posterior distribution. At each step a sample is drawn from the posterior through which gradients can flow and the ELBO is evaluated. This can be backpropagated through to get gradients with respect to $\phi$ and $\theta$ which can then be updated as appropriate using an optimiser of the practitioner's choice.

## 5 CASE STUDY: LIVER TRANSPLANTATION ACCEPTANCE DECISIONS

In this section we explore the explainability benefits of our method for evaluating real decision making. Given space constraints, we relegate validation of the method on synthetic examples to the appendix, and focus here on a real medical example of accepting liver donation offers for transplantation. This is an important area where much has been done to improve decision support for clinicians (Volk et al., 2015; Berrevoets et al., 2020), but which often focuses on simply suggesting theoretically optimal actions. Here, by trying to understand better the decision making process, we hope to be able to provide more bespoke support in the future.

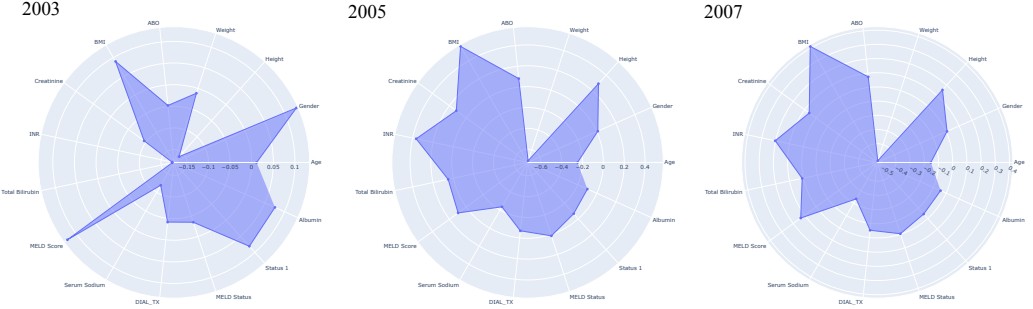

Figure 5: **Changing Focuses.** Predicted weights of different covariates at two-year time intervals.

Organ transplantation is an extreme medical procedure, often considered the last resort for critically ill patients (Merion et al., 2005). Despite the significant risks associated with major surgery, transplantation can offer the chance of a relatively normal life (Starzl et al., 1982) and as a result demand for donor organs far outstrips supply (Wynn & Alexander, 2011). Yet, perhaps surprisingly, a large proportion of organ offers are turned down by clinicians (in consultation with the receiving patients) for various reasons including donor quality or perceived poor compatibility with the recipient (Volk et al., 2015). This decision faced by clinicians can be (under some simplification) described by the process: 1) A donor organ becomes available and is offered to a patient; 2) The clinician considers the state of the organ, the state of the patient, as well as other information such as the previous reject history of the organ (this is the context $X$); 3) The clinician makes the binary decision to accept/reject the organ (the intervention $A$); 4) The patient receives/does not receive the organ and a measure of their outcome is recorded - we measure their survival time as the main outcome of interest (the outcome $Y$). Full details of the dataset, preprocessing steps, and experimental setup can be found in the appendix. Note that in this example we are not trying to capture the policy of any individual doctor, but rather the general policy of a treatment centre over time and how what they find important changes - this is due to the form of the data available, but the method could equally be applied to individual clinicians if that information was made available. Additionally, while survival time will not be recorded immediately, the most powerful signal in this case is usually a death that follows very soon after the decision, which will be effectively captured in the data.

**Evolving Relative Importances.** First, we now examine exactly how our method can explain a non-stationary policy over time. In Figure 5 we plot the relative weights for covariates in terms of the treatment effect for accepting the organ offer as they change at two-year periods. The Model for End-Stage Liver Disease (MELD) score (Kamath & Kim, 2007) was introduced in 2002 to give an overall rating of how sick a patient is. On introduction, it was the main consideration when making an acceptance decision, as can be seen clearly in Figure 5. However, over time this became seen as less important as clinicians started to not want to rely on a single score, with some clinicians even turning down offers for patients with high scores in hopes of a better match, with one study showing that 84

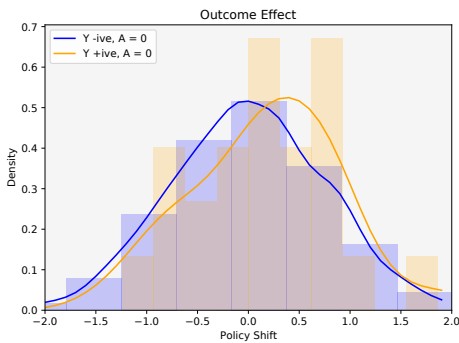

Figure 6: One-step policy shift densities after observing an outcome.

% of the patients who died while waiting for an organ had MELD scores $\geq 15$ and had previously declined at least one offer (Lai et al., 2012). This can also be seen reflected in Figure 5 as the importance of the MELD score in accepting the offer decreases and the policy stabilises.

**Reactionary Policies** The most important aspect of our method is that we can model the reaction of the agent to their experience on the fly. For example we can see how observing various outcomes change the opinion of the agent moving forward. This is demonstrated in Figure 6 which plots density estimates of the *policy shift* (measured as the distance the decision boundary moves) after rejecting the offer and measuring a positive/negative outcome. Here it can be

seen that when an offer is rejected then a negative outcome shifts the decision boundary downwards more than a positive outcome, making it less likely to reject an offer in the next time-step.

**Matching Clinical Decisions.** While it should be emphasised that our method is (to the best of our knowledge) the *only* method with the aim or capability to model the evolving decision process of the agent, as mentioned in Section 3 there are a variety of methods that are tangentially related. First, we consider a few standard approaches to imitation learning with *behavioural cloning* (BC) (Bain & Sammut, 1995), where we consider both a linear (**BC-Linear**) and deep (**BC-Deep**) function-approximator, representing both ends of the interpretability spectrum for BC. Continuing in imitation learning, we also compare alongside *reward-regularised classification for apprenticeship learning* (**RCAL**) (Piot et al., 2014), a method that regularises the reward implied by the learnt $Q$-values. We additionally consider an adaptation of *counterfactual inverse reinforcement learning* (**CIRL**) (Bica et al., 2021) to the bandit setting where we first estimate the true CATE of the interventions and thus the true potential outcomes. We use these as the reward and apply an optimal policy, which in this one-step case is trivially achieved by choosing the action that maximises the potential outcome.

In Table 2 we compare the predictive power of our method and all benchmarks on the task of imitation learning – i.e. matching the actions of the real demonstrator in some held-out test set. This is the only task for which we can make meaningful comparison with the existing literature, given our divergent goals. Despite the fact that at each step our method only considers a linear decision boundary, we are still able to outperform the deep network architectures used in BC and RCAL in terms of AUC and APS. This can be put down to the flexible way in which the policy can change over time to allow for adaptation. Less surprisingly, we also outperform the stationary linear classifier as well as the version of CIRL, the poor performance of which is probably explained by the unrealistic nature of the causal assumptions in this setting. Further experimental results on action matching in on multiple additional datasets are provided in the appendix.

Table 2: **Action prediction performance.** Comparison of methods on transplant offer acceptances. Performance of the methods is evaluated on the quality of *action matching* against a held out test set of demonstrations. We report the area under the receiving operator characteristic curve (AUC) and the average precision score (APS).

| Method | AUC ↑ | APS ↑ |
|---|---|---|
| BC-Linear | $0.798 \pm 0.002$ | $0.647 \pm 0.001$ |
| BC-Deep | $0.803 \pm 0.002$ | $0.655 \pm 0.001$ |
| RCAL | $0.797 \pm 0.008$ | $0.662 \pm 0.003$ |
| CIRL | $0.553 \pm 0.008$ | $0.510 \pm 0.002$ |
| IOL **(Ours)** | $0.824 \pm 0.006$ | $0.677 \pm 0.011$ |

## 6 DISCUSSION

**Limitations.** By employing a deep architecture our method is able to learn a non-parametric and accurate form of both the non-stationary policy and update function, making far fewer assumptions on their true form than is normal in the literature. This does mean, however, that it works best when training data is abundant, and can otherwise be prone to over-fitting as is common with deep neural networks. Additionally, the assumption that agents are always taking the action they believe will maximise the outcome does mean our method is out-of-the-box unable to account for strategies that deliberately pick actions that could be sub-optimal but may yield useful information (i.e. exploratory strategies). An interesting future direction would include exploring this, potentially by augmenting the outcome to include a measure of expected information.

**Societal Impact and Ethics.** Being able to better understand decision making processes could be highly useful in combating biases or correcting mistakes made by human decision-makers. Nonetheless, it should be noted that any method that aims to analyse and explain observed behaviour has the potential to be misused, for example to unfairly attribute incorrect intentions to someone. As such, it is important to emphasise that, as is the nature of inverse problems, we are only giving *one* plausible explanation for the behaviour of an agent and cannot suggest that this is exactly what has gone on in their mind.

**Conclusions.** In this paper we have tackled the problem of *inverse* online learning – that is descriptively modelling the process through which an observed agent adapts their policy over time – by interpreting actions as targeted interventions, or treatments. We drew together policy learning and treatment effect estimation to present a practical algorithm for solving this problem in an interpretable and meaningful manner, before demonstrating how this could be using in the medical domain to analyse the decisions of clinicians when choosing whether to accept offers for transplantation.

ACKNOWLEDGEMENTS

AJC would like to acknowledge and thank Microsoft Research for its support through its PhD Scholarship Program with the EPSRC. AC gratefully acknowledges funding from AstraZeneca. This work was additionally supported by the Office of Naval Research (ONR) and the NSF (Grant number: 1722516). We would like to thank all of the anonymous reviewers on OpenReview, alongside the many members of the van der Schaar lab, for their input, comments, and suggestions at various stages that have ultimately improved the manuscript.

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

APPENDIX

---

**Algorithm 1:** Inverse Online Learning (IOL)

---

**Result:** Parameters $\phi$ of variational distribution and $\theta$ of generative model
**Input:** $\mathcal{D}, \mathcal{X}, \mathcal{A}, \mathcal{Y}$, learning rate $\eta$;
Initialise $\phi, \theta$;
**while** *not converged* **do**
    Evaluate $p_\theta(m_t|m_{t-1})$ ;    ▷ Calculate forward model next step priors
    Evaluate $q_\phi(m_t|m_{t-1}, \vec{h})$ ;    ▷ Inference network to get posterior
    Sample $\hat{\vec{m}} \sim q_\phi(\vec{m}|\vec{h})$ ;    ▷ Straight-through MC sample
    $NLL_{loss} = \mathbb{E}_{q_\phi(\vec{m}|\vec{h})}\big[p_\theta(\vec{a}|\vec{m})\big]$;
    $KL_{loss} = D_{KL}\big(q_\phi(m_1|\vec{x})||p_\theta(m_1)\big) + \sum_{t=2}^{T} \mathbb{E}\big[D_{KL}\big(q_\phi(m_t|m_{t-1}, \vec{h})||p_\theta(m_t|m_{t-1})\big)\big]$;
    $\mathcal{F}(\phi, \theta) = NLL_{loss} + KL_{loss}$;
    $(\phi', \theta') \leftarrow (\phi, \theta) - \eta\nabla_{\phi,\theta}\mathcal{F}(\phi, \theta)$ ;    ▷ Gradient step for $\phi, \theta$
    $\phi, \theta \leftarrow \phi', \theta'$
**end**
**Return:** $\phi, \theta$

---

## A   EXPERIMENTAL DETAILS

All experiments were performed on a 2016 MacBook Pro, using a 2.9 GHz Dual-Core Intel Core i5 with 8GB of LPDDR3 RAM and no GPU acceleration.

Code was written in PyTorch Paszke et al. (2019) and no external code was used in the implementation of either the model nor the benchmarks. Hyperparameters were selected through grid search over a validation fold of the training data.

All medical data used has undergone a de-identification process and consent was obtained by the relevant curators for their data to be publicly released.

### A.1   UNOS DATA

Our data focuses on data from the United Network for Organ Sharing (UNOS), the US non-profit that manages their national waiting list and maintains a database and record of transplantations that occur.

We focus on liver-transplantations, where the donor is deceased. The data is made up of patient details from various centres around the United States. Of the multiple million records of individual transplantations available, once those with missing information on either the donor or recipient are removed we are left with around 500,000 patients spread across 31 centres. Available variables are listed in Table 3

In our action matching experiments we split the data into Train/Validation/Test folds by separating the data by centre. In particular we use centre CTR23901 with 114,314 patients for training; CTR124 with 21,067 patients for validation; and finally CTR279 with 14,295 patients for testing.

For the more exploratory experiments who's focus is on inference over some data and not extrapolated prediction, we simply train on centre CTR23901, and examine what is inferred about the treatment of those specific patients

## B   ADDITIONAL VALIDATION

While we focused on an important case study in the main paper, we appreciate the need to provide additional evidence that our model can work outside of one specific case example. As such, we now report results on some experiments focusing first on a toy (but relevant) synthetic example, followed by performance results on multiple further real datasets.

Table 3: **UNOS Variables**

| | |
|---|---|
| AGE | Recipient's age at transplant |
| AGE_DON | Donor's age at transplant |
| ALBUMIN_TX | Recipient's albumin concentration at transplant |
| ASCITES_TX | Recipient had ascites at the time of transplant |
| BMI_CALC | Recipient's BMI |
| COD_CAD_DON | Cause of death of the donor |
| COLD_ISCH | Cold ischemia time - length of time from the donor being recovered until it is transplanted |
| CREAT_DON | Donor's creatinine concentration |
| CREAT_TX | Recipient's creatinine concentration at transplant |
| DIAL_TX | Whether recipient was on dialysis before transplant |
| ETHCAT | Recipient's ethnicity |
| ETHCAT_DON | Donor's ethnicity |
| EXC_HCC | Whether the recipient received a hepatocellular carcinoma exception point |
| FUNC_STAT_TRR | Recipient's functional status at the time of transplant |
| HCV_SEROSTATUS | Whether the recipient has hepatitis C |
| INR_TX | Recipient's INR value at the time transplant |
| LIFE_SUP_TRR | Whether the recipient was on life support at the time of transplant |
| MED_COND_TRR | Whether the recipient was in an icu, hospital bed, or came from home before transplant |
| MELD_PELD_LAB_SCORE | Recipient MELD score |
| NON_HRT_DON | Whether the donor was a DCD (donation after cardiac death) |
| NUM_PREV_TX | Number of previous liver transplants the recipient has had |
| ON_VENT_TRR | Whether the recipient was on a mechanical ventilator before transplant |
| PORTAL_VEIN_TRR | Whether the recipient had a portal vein thrombosis at the time of transplant |
| PREV_AB_SURG_TRR | Whether the recipient had prior abdominal surgery before the liver transplant |
| PREV_TX | Whether the recipient had any previous transplant |
| SHARE_TY | Whether the donor is local, regional, or national |
| TBILI_TX | Recipient's bilirubin at the time of transplant |
| TXLIV | Whether the graft is whole or segmental |
| death_mech_don_group | Donor mechanism of death |
| deathcirc | Donor circumstance of death |
| diag1 | Recipient's primary cause of liver disease |
| meldstat | Whether the recipient is status 1a |
| statushcc | Whether the recipient has hcc |
| status1 | Whether the recipient is status 1a |

## B.1 SYNTHETIC EXPERIMENT

We consider a toy example of an agent that is learning online about the treatment effect of the interventions they are performing.

We let $\mathcal{X} = \mathbb{R}^5, \mathcal{A} = [0,1],$ and $\mathcal{Y} = \mathbb{R}$, with the true outcomes a linear combination of the features given the action (with weights stochastically samples). From a sampled prior set of weights in the agents mind, the agent then interacts, estimating the potential outcomes and choosing an action stochastically according to equation 7 with $\alpha = 1, \beta = 0$. After every context, action, outcome triple $(x, a, y)$, the agent adjusts it's beliefs over the weights of the potential outcome functions using an online gradient descent method:

$$\omega_a = \omega_a - \lambda((\tilde{y} - y) \times x), \tag{11}$$

with $\lambda$ some learning rate. Clearly then the policy of the agent is non-stationary and changes over time in reaction to the observed outcomes.

**Results.** We simulate 10000 trajectories of length 50. When applying IOL to this simulated data, we can achieve an accuracy of 99%, meaning 99% of the time we can accurately determine which of the potential outcomes the agent thins will be higher. This is compared to a stationary estimate of the CATE which can only reach 52% given the fact that the policy is detached from the truth of the treatment effect. Clearly this is an extreme example designed to fail the existing approaches but it does serve to show how stark the contrast is.

## B.2 FURTHER PERFORMANCE ON REAL DATA

Alongside the synthetic example above, we also provide further results on real medical data, demonstrating improved performance across a variety of domains. These datasets do not contain a recorded outcome, consequently so we do not include CIRL, and the next state is used as a substitute in IOL. Results are given in Table 2, and a description of the datasets follows:

**Intensive Care Unit.** The *ICU* data covers the treatment of 23,106 of patients in the intensive care unit from Amsterdam UMC Elbers (2019). These patients are in a more critical state than those on the general wards while suffering similarly from a variety of conditions and consequently are monitored more frequently, with the database containing around 1 billion clinical observations at varying timesteps down to minute by minute recordings. The data is aggregated into one hour timesteps and models the treatment of the prescription of antibiotics. Included are 24 series features focusing on vital signs including heart rate, blood pressure, and various chemical blood concentration levels.

**Cystic Fibrosis.** The *CF* data considers patients enrolled in the UK Cystic Fibrosis Registry Taylor-Robinson et al. (2018), coviering around 5,800 patients over the course of multiple years. Measurements were taken on average every 6 months, and the 79 features cover common comorbidities, test results, and infections that arise. The intervention corresponds to the prescription of cortico steroids.

**Hospital Wards.** The *Wards* data is based on the care of 6,321 patients at the Ronald Reagan UCLA Medical Center in California who were treated on the general medicine floor between 2013-2016 Alaa et al. (2017). These patients were treated for a variety of conditions including pneumonia, sepsis, and fevers were in general stable and deterioration that required ICU care was rare. Measurements were taken roughly every 4 hours, with average stays lasting 9 days, and include common vital signs such as pulse and blood pressure alongside lab tests and results and 35 temporal features. The action space is taken as a choice of application of oxygen therapy.

Table 4: **Medical examples performance.** Comparison of methods on three different medical datasets. Performance of the policy is evaluated on the quality of *action matching* against a held out test set of demonstrations. We report the area under the receiving operator characteristic curve (AUC) and average precision score (APS).

| | ICU | | CF | | Wards | |
|---|---|---|---|---|---|---|
| Metric | AUC | APS | AUC | APS | AUC | APS |
| BC-L | $0.681 \pm 0.001$ | $\mathbf{0.655 \pm 0.000}$ | $0.830 \pm 0.010$ | $0.797 \pm 0.009$ | $0.896 \pm 0.006$ | $0.853 \pm 0.002$ |
| BC-D | $0.668 \pm 0.003$ | $0.654 \pm 0.001$ | $\mathbf{0.952 \pm 0.010}$ | $0.834 \pm 0.003$ | $0.845 \pm 0.010$ | $0.811 \pm 0.013$ |
| RCAL | $0.396 \pm 0.018$ | $0.563 \pm 0.007$ | $0.689 \pm 0.022$ | $0.745 \pm 0.008$ | $0.942 \pm 0.003$ | $0.920 \pm 0.004$ |
| IOL | $\mathbf{0.686 \pm 0.092}$ | $0.631 \pm 0.065$ | $0.942 \pm 0.031$ | $\mathbf{0.943 \pm 0.026}$ | $\mathbf{0.965 \pm 0.031}$ | $\mathbf{0.941 \pm 0.055}$ |

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
