# OpenReview forum: "Inverse Online Learning: Understanding Non-Stationary and Reactionary Policies"
_ICLR.cc/2022/Conference — ICLR 2022 Poster_

### Official Review · Reviewer_S6Ms · 2021-10-31

**Correctness:** 3
**Technical Novelty And Significance:** 3
**Empirical Novelty And Significance:** 3
**Recommendation:** 6
**Confidence:** 3

**Main Review:**

Strengths:
1. The framework is a principled approach to the analysis the authors wish to perform in the organ exchange setting (where the linear restriction seems to fit well—maybe? see discussion).
2. The performance of the model "looks good" in the sense that it produces plausible explanations and has a better AUC than the baselines.
3. The authors seem to be the first to identify this problem as of interest (see discussion about novelty in weaknesses).
4. The authors' linear decision-making approach with linear weights might make their approach compatible with linear statistical approaches that are compatible with the medical community.

Weaknesses:
1. In my opinion, the authors' claims about novelty are somewhat overstated. For example, for IRL:
- Many IRL methods can produce an imitative policy as well as a reward estimate (e.g., Ho and Ermon (which the authors cite) and Fu, Luo and Levine). Particularly relevant to the authors' response is section 7.3 of Fu, Luo and Levine where they evaluate on pure imitation learning tasks, finding performance that is comparable with pure imitation learning methods.
- IRL can produce non-stationary policies if the state includes the appropriate observations that would induce non-stationarity. In the authors' case, if the history of (X,A,Y) up to time t is included in the state at time T, the policy produced can be non-stationary in the way the authors desire. This is not that different from what the authors end up doing.
- If the requirement is that a linear decision policy is learned, this can be accomplished by having the neural network produce the action weights as output and then using them as input to the dot product. This is essentially what the authors end up doing.
2. There are many different design decisions about how this procedure could be carried out—from more macro-level decisions like whether the memory should be stochastic or deterministic, whether to use an RNN-type structure for memory (as the authors have done) vs. a GRU or LSTM, to micro-decisions like hyperparameter selection, whether to use regularization, etc. There isn't a lot of justification that appears in the paper, which makes it hard for subsequent authors to know whether the authors tried other ideas and they failed or if this was just the first pass. The authors perform no ablation testing on model features.
3. I am somewhat concerned about the limitations of the linear model. There are many simple decision rules that cannot be captured in a linear model—for example, rejecting a patient if a particular feature is too high/low. I think the authors are a little glib about what linear models can cover.

Post-review: I have lowered my review slightly after reading the other reviews. I thank the authors for their response.

**Summary Of The Paper:**

The objective of the paper is an interpretable descriptive model of online learning from trajectories of the form $(x_t,a_t,y_t)$, where $x_t$ is an i.i.d. context, $a_t$ is the action taken that depends on the context and history $(x,a,y)$ up to time $t-1$, and $y_t$ is the outcome resulting from the application of $a_t$ to $x_t$.

The authors' motivating example is liver donation acceptance decisions: a potential donor approaches with an organ they are offering to a particular recipient. The treatment center makes a binary decision to accept or reject the donation based on features of the donor and the recipient. The outcome is the survival time of the patient.

The structure of the authors' model is based on three components:
1. A memory summarization network that maps from the history up to time $t-1$ to the parameters of a multivariate normal distribution over memory vectors. The specific structure is that a new memory vector is produced from the previous memory vector and the observation from the previous time step.
2. An outcome estimation network. This network takes the memory vector as input and produces weights for each action. A dot product of these weights plus the (exogenous) action features yields the agent's estimate of the outcome of each action.
3. The treatment rule is just a logistic function applied to the predicted outcomes.

The model is trained using stochastic variation inference with the ELBO using an LSTM.

Empirically, the authors find that their approach outperforms action prediction baselines such as behavioral cloning, reward-regularised classification for apprenticeship learning and counterfactual inverse reinforcement learning. Additionally, they use their model to analyze shifts in the transplant policy over time, e.g., the declining importance of MELD score and how offers that are rejected make it less likely than a subsequent offer will be rejected.

**Summary Of The Review:**

The paper identifies an interesting problem that takes a different perspective than past work. It covers a lot of related literature but is a bit tunnel-visioned on the problem's perceived newness. The algorithmic approach introduced is a pretty basic tweak on existing methods, but the performance is promising.

---

> ### Author Response · Authors · 2021-11-16
> **Author response to Reviewer S6Ms**
>
> Thank you very much for your thoughtful review and helpful comments! Please find below some responses to some specific comments of you review that we think may be helpful to clarify our position:
>
> “In my opinion, the authors' claims about novelty are somewhat overstated…” Thank you for your comment - we agree that it would be wrong to suggest that IRL methods can’t be used to obtain a policy, and will clarify this in the paper. Of course they can if, having recovered the reward signal, you run an appropriate RL method to obtain an optimal policy w.r.t the recovered reward. Our point is simply though that just the IRL procedure does not itself produce a policy as it focuses on obtaining the reward function - with that in mind we don’t consider methods like GAIL (Ho & Ermon 2016) to actually be true IRL methods, since they never actually calculate an explicit reward function. The point of that work is to show that the composition of IRL and RL can be done directly, with the IRL step done **implicitly**. As such, we would consider them simply as IL methods - those that directly learn an imitator policy from the demonstrations.
>
> “IRL can produce non-stationary policies…” We agree that it is certainly true that if you augment the statespace appropriately then standard IRL methods could be used to obtain a non-stationary policy. This is mostly a product of the fact that an MDP is an incredibly general framework, and similar to how you can express any POMDP to an MDP where the statespace is augmented to be all possible beliefs over underlying states. Of course though, you then run into issues when applying standard MDP techniques (normal RL/IRL methods), that while theoretically possible, practically don’t work out well - in most part due to the dimensionality of the statespace exploding.
>
> “There are many different design decisions about how this procedure could be carried out…”
> Thank you for pointing this out - of course you are completely correct to suggest there are many parts of the described model that can be replaced including swapping the exact recurrent architecture used in both the memory and inference networks. In the paper, we would consider the exact architecture to be of secondary importance, the main point being the principal idea to use this kind of model for understanding the agent’s evolving policy, but we appreciate the need to clearly express what has been considered. To that end, we will include a discussion on hyper-parameter (including architectures) selection and will look to add an ablation study in the appendix.
>
> “I am somewhat concerned about the limitations of the linear model…”
> It is true that linear models are restricted in the class of policy that they are able to model. We believe though that they are a reasonable tradeoff between expressibility and interpretability in this case where they are free to evolve over time, and as you mention, they are very standard especially within the medical community. With all parameterised policies, there will have to be some approximation made and the policies you describe could be reasonably well approximated through having large weights associated with that variable and a set constant offset. Given prior knowledge though, it would also transform some of the variables of interest into appropriate indicator variables for a given threshold.
>
> Once again, thank you for your review, we hope our response has further justified some aspects of the paper and we look forward to hearing back from you!

---

### Official Review · Reviewer_F1MZ · 2021-11-01

**Correctness:** 3
**Technical Novelty And Significance:** 3
**Empirical Novelty And Significance:** 3
**Recommendation:** 8
**Confidence:** 3

**Main Review:**

This is an interesting paper that presents a new and interesting problem. I think the use of generative modeling and inference to gain insight into decision making of an online learning agent is quite innovative. Overall, I think it is an interesting paper. However, I felt some part of the paper could be clearer.
Example: CATE: conditional average treatment effect. It is not clear if the paper is introducing this terms or borrowing it from existing literature because there is no citation.
Also, it seems weird that the perceived utilities of an agent’s decision is given what seems to be a close to a medical term?

Assumption 2 is not clear at all. The first part of assumption2 is clear but the second part
“Given a stochastic policy,  t(x) is a monotonic increasing function of the agent’s belief over the
treatment effect ~  (x; t).
”
does not seem to be complete statement. What is monotonic?

I felt the experiments in the paper were quite weak. The paper shows that the proposed algorithm outperforms existing imitation learning methods in terms of predictive power. However, what would also have been interesting is to see success/failure of existing imitation learning methods for gaining insight into decision making process of organ donation acceptance decision. Is that possible? If no, why not and if yes, what is the motivation behind the new method.

One of the claims of the paper is to formulate the problem of inverse online learning. While after reading the paper I can guess what the formulation might be. However, I dont think that the paper states the formulation explicitly. As in given the data, we would like to find X.






**Summary Of The Paper:**


This paper proposes an approach for understanding the decision making of a black-box online learning agent from the a dataset of observed trajectories. Given a dataset of trajectories / decisions made by an agent along with initial context and resulting outcome, the paper proposes a method to perform inference over the generative model of the given dataset using variations inference. The results are used to interpret the decision making of an agent that is learning online. The paper demonstrates its usefulness by highlighting insights into factors that may govern decision on a organ donation acceptance decision dataset.

**Summary Of The Review:**

Overall, I like the paper but it can improve on clarity and experiments.

---

> ### Author Response · Authors · 2021-11-16
> **Author response to Reviewer F1MZ**
>
> Thank you very much for your thoughtful review and helpful comments, we are glad that you found the paper and problem interesting, and will aim to clarify the points you raised in your review, in particular:
>
> “Example: CATE: conditional average treatment effect...” Thank you for your point, we will clarify the origin of CATE in the paper. While it is a term that is widely used in the treatment-effect literature, we appreciate the definition and use of the term may not be obvious to many readers. We chose to use it as it is widely used in the medical, but also importantly economic, literature - providing useful interpretive benefit to the method. We think that describing policies in terms of the expected impact that the actions will have is a natural way to think about why an agent will have made a certain decision, although of course we appreciate that it is not the only possible way.
>
> “Assumption 2 is not clear…”
> The second part of Assumption 2 is simply stating that the policy (which is the probability of taking the action) is a monotonic increasing function of the treatment effect. I.e if the treatment effect is larger then the probability of treatment cannot be lower. This seems very reasonable - if the treatment effect is higher, there shouldn’t really be any reason for the action to be less likely. In order to make this clearer, we will amend Assumption 2 to read “Given a stochastic *agent*, the policy $\pi_t(x) = P(A_t =1 |x)$ is a monotonic increasing function…”.
>
> “what would also have been interesting is to see success/failure of existing imitation learning methods…”
> Thank you for these useful suggestions, we can certainly include some failure cases of existing methods in the paper to make it clearer the need for methods like ours. Essentially, the fact that they do not model a non-stationary policy makes them incapable of modelling the kind of policies we are aiming for. The effect of this can be most obviously seen when compared to Figure 5 in the paper, those methods will only produce a single plot over the entire trajectory, not allowing us to see a changing policy. We will include a demonstration of this in the main paper, overlaid with the policy that we learn.
>
> “I dont think that the paper states the formulation explicitly…”
> Thank you for pointing out that this is not explicitly said - we will make this clearer in the paper. For avoidance of doubt, the goal is, having seen the data, to determine the evolving policy and beliefs of the agent as it changes over time.
>
> Once again, thank you for your review, we hope our response has clarified some points of the paper and look forward to hearing back from you!

---

> > ### Comment · Reviewer_F1MZ · 2021-11-28
> > **thanks**
> >
> > Thanks for the response. I believe with the improvements pointed in the response the paper will improve further.

---

### Official Review · Reviewer_vs7Z · 2021-11-02

**Correctness:** 2
**Technical Novelty And Significance:** 2
**Empirical Novelty And Significance:** 2
**Recommendation:** 6
**Confidence:** 3

**Main Review:**

1. The setting and description is quite confusing. The goal as stated in Sec 2 seems to suggest the problem is about "inverse reinforcement learning" while also stating the goal is to learn a non-stationary policy from trajectories as in imitation learning under a causal framework.

2. The paper overall sounds like that of learning non-stationary policies under "misspecification" of the counterfactual response functions (linear in covariates/confounders), which is almost inevitable if we go by a parametric way of solving these problems, and that I believe is a much more transparent setup to motivate the problem. But the whole narrative about "inverse learning" is extremely confusing and in my opinion redundant. If I am missing something, I would ask authors to clarify. The narrative also has the danger of now confusing IRL as a task where the explicit goal is to summarize beliefs over the rewards and policy learning itself.

3. The optimization and inference is not super well motivated. Why does maximizing likelihood the only and appropriate goal here? It makes sense, because the assumption on how the agent is acting based on the current estimates of response functions. In that sense I don't disagree but I believe the presentation needs to be significantly improved and the cost well motivated. Right now the motivation section and inference seems very disconnected.

4. Also validation using synthetic example seems insufficient. Comparison to a stationary assumption is only one baseline. Isn't it important to also focus on gradient updates being noisy due to the model mis-specification of the response functions? Also the more interesting evaluation seems to be critical to show to over time.

5. Table 2 results are confusing especially comparison to CIRL, what are the "unrealistic" assumptions CIRL is making that is being relaxed in this work? In fact it seems a bit weird to me that CIRL and IOL are not performing comparably modulo non-stationarity. If that is the main issue, (and I am not sure because i am not familiar with CIRL), then it doesn't seem to be about causal assumptions. Can authors please clarify their empirical evaluation more clearly?

**Summary Of The Paper:**

The paper motivates an "inverse online learning" framework to learn non-stationary policy online. The generative assumption is a deep state-space model and it is assumed that the agent acts optimally according to the current belief in potential outcomes and updates its model based on the true outcomes observed. The critical assumption seems to be that of non-stationarity over the policy function as it responds/interacts with the environment based on updated estimates of its belief over the response function. The parameters of the deep state space model are estimated using the ELBO over the log likelihood of the observed trajectories. Synthetic and real-world validation demonstrates some benefit in terms of "identifying the optimal treatment" based on the true model when acting based off the estimates.

**Summary Of The Review:**

Overall I believe the presentation of the technical problem is confusing, could be significantly more clear, cost function/inference better connected to the problem and validation improved. I strongly believe the validation should include some form of real misspecification and then evaluate how the learned policy improves over time.


Post-discussion update - Thank you to the authors for their clarifications. I have updated my score based on their response.

---

> ### Author Response · Authors · 2021-11-16
> **Author response to Reviewer vs7Z**
>
> Thank you very much for your review and thoughtful comments.
>
> Before we address some of the specific points mentioned in your review we would like to start by clearing up a potential misunderstanding that may have arisen based on the summary you wrote of the paper where you say “Synthetic and real-world validation demonstrates some benefit in terms of "identifying the optimal treatment" based on the true model when acting based off the estimates.”. We should make it clear that our goal is very much **not** "identifying the optimal treatment", but rather to understand how/why an agent came to the decision to select the treatment that they did, whether it was optimal or not. In the context of Table 1, this corresponds to learning the non-stationary observational policy $\pi_{t}^{obs}$, as opposed to an optimal policy $\pi_{t}^{opt}$. This is an important difference in goal and motivates a lot of the design choices of our presented algorithm.
>
> “The setting and description is quite confusing...” We will aim to re-word section 2 to make it clearer as to exactly what setting we are covering. Specifically the last two sentences of the first paragraph are not designed to suggest that what we are doing has the same goal as both of these frameworks, for it is not. Merely we are saying that these two areas are the most closely related. In summary, the setting we are considering is one where we watch an agent interact with an environment while *they (not us) learn online* - based on these demonstrations we would *post hoc* like to understand how their beliefs evolved over the course of their interaction with the environment and why they took the actions they did when they did. This relates to IRL in it's goal of uncovering a driving force (the reward in IRL) of a demonstrator, but also given our Potential Outcomes framing and policy learning also relates to causal imitation learning.
>
> “The paper overall sounds like that of learning non-stationary policies…”
> We would like to clarify that we are not learning under any misspecification of the response surfaces. Rather we are modelling an agent who *we assume has themselves potentially misspecified* the response surface, and that they may change their perceived response surface over the time period. As part of this we want to learn *how* the agent has misspecified the response surfaces and requires us to focus on them, not the true response surfaces. With this clarification we hope it is clearer why the inverse learning narratives appropriate, please let us know if it is still unclear!
>
> “The optimization and inference is not super well motivated…”
> Our main goal is to learn the beliefs of the agent over the seen trajectory, in this respect our goal is not actually just to maximise the likelihood. Instead, having seen the data, we are trying to reason about the posterior distribution over the beliefs at every timestep - this Bayesian approach is very natural, theoretically motivated, and widely accepted. As it happens, we can’t evaluate the posterior exactly, and so we resort to variational inference to maximise an ELBO - which is again theoretically motivated and widely accepted. We will aim to clarify this fully in the paper - especially to make clear that the goal is to reason about the posterior distribution as we hope that should make the need for the ELBO objective function clear.
>
> “Also validation using synthetic example seems insufficient…”
> We’re not sure exactly what you mean by “focus on gradient updates being noisy due to the model mis-specification of the response functions”, perhaps though this is related to the point we made earlier in this response, that we are not considering learning under our own misspecification of the response function, but rather that we consider the agent to have potentially done so, which we would now like to model. Indeed we agree that the evaluation over time is the interesting part, and we aim to show how this can be useful already in the paper on the real data we use (for example in Figure 5 ). We include the synthetic data in the appendix for completeness.
>
> “Table 2 results are confusing especially comparison to CIRL…”
> The main, and *critical*, difference between CIRL and IOL is that in CIRL it is assumed that the agent knows the **true** response surfaces, while IOL assumes that the agent may have an **incorrect** and **non-stationary** belief over what the response surfaces look like. We think it’s very unlikely that the agent will know the true response surfaces, so trying to learn an imitation policy under those assumptions will likely not be able to perfectly capture their behaviour. On the other hand, by flexibly assuming that they may not have correct beliefs (and that they may change) we should be able to match their policy more closely.
>
> Once again, thank you for your review, we hope our response has clarified some of you concerns and look forward to hearing back from you!

---

### Decision · Program_Chairs · 2022-01-20

**Decision:**

Accept (Poster)

**Comment:**

All three reviewers suggest acceptance of the paper. The authors study an interesting problem (understanding non-stationary and reactionary policies) and propose a solution to the problem which compares favorably to baselines in experiments. However, some of the reviewers also criticize unclarities in the presentation of the paper and the made assumptions. The authors clarified those points quite well in their rebuttal. Further concerns regarded design decisions and the comparison to failure cases of baselines. The authors addressed those in their rebuttal and promised to include corresponding material in their updated paper. Hence I am suggesting acceptance of the paper. Nevertheless, I would like to urge the authors to carefuly revise their problem presentation in the paper in order to improve clarity and add the promised additional insights to the final version of the paper.